# Development of test bench to determine the distribution of granular fertilizers in planting rows using spiral roller, two spiral rollers and fluted roller

**Gabriel Ganancini Zimmermann** [1]*, **Samir Paulo Jasper**[1], **Leonardo Leônidas Kmiecik**[1], **Lauro Strapasson Neto**[2], **Thiago Xavier da Silva**[2], **Yasser Alabi Oiole**[1]

**1** Soil and Engineering Department Agricultural, Graduate Program in Soil Science, Agrarian Sciences Sector, Federal University of Paraná, Curitiba, Paraná, Brazil, **2** Agronomy, Gearing Farm Tractor Laboratory, Agrarian Sciences Sector, Federal University of Paraná, Curitiba, Paraná, Brazil

* gabrielganancini@gmail.com

**Data Availability Statement:** All relevant data are within the paper and its Supporting information files.

## Abstract

The success of the application of granular fertilizers (GFs) in planting rows depends on the uniformity and performance of product dispensing systems, which are influenced by external factors. The objective of this study was to determine the outflow rates of two GF formulations ($GF_1$ 04-14-08 and $GF_2$ 04-30-10) using three types of fertilizer spreader—with one spiral roller (A), two spiral rollers (B), or a fluted roller (C)—and three operating speeds (1,11, 1.94, and 2.77 m s$^{-1}$). The following parameters were determined in GFs: density, angle of repose, water content, and segregation (particle size). In the designed test bench, GFs were transferred from a reservoir to a spreader, and ultimately to a container, where they were weighed, and data were transmitted to the data acquisition system (DAS). A total of 7,560 outflow data points were collected (g s$^{-1}$) and subjected to descriptive analysis of measures of central tendency, dispersion, asymmetry, and kurtosis, and Shewhart control charts were generated. Particle density and segregation were significantly different between the GFs, whereas the angle of repose and water content were not significantly different. The bench design and the DAS allowed measuring the outflow of GFs in different spreaders and demonstrated that this parameter was influenced by particle segregation. The segregation of $GF_1$ was higher than that of $GF_2$. The outflow variability at the speed of 1.11 m s$^{-1}$ was lower, and the spreader with a fluted roller had the highest uniformity and was the most suitable for application with variable rates.

## Introduction

In recent years, agricultural production in Brazil has increased significantly with the adoption of new technologies. Increased production is primarily attributed to improvements in agricultural technologies and management practices, including the use of direct sowing techniques and genetic improvement sit [1]. In this scenario, the demand for granular fertilizers (GFs)

**Funding:** The authors received no specific funding for this work.

**Competing interests:** The authors have declared that no competing interests exist.

follows the same trend, especially for phosphate fertilizers in low-fertility tropical soils to improve agricultural production sit [2]. For this reason, GFs are applied in planting rows using uniform dispensing mechanisms in simpler production systems. In contrast, variable rate technology allows streamlining fertilizer application and utilization, which is essential for modern precision agriculture sit [3]. The distribution of GFs depends mainly on the quality of dispensing systems, which are affected by external factors sit [4]. sit [5] developed mathematical models to measure application rates of granulated fertilizers at different dosing mechanisms (single helical), as well as speed and longitudinal and transverse slopes. The authors observed rate differences among the evaluated spreaders, with greater deposition variability if under varying inclinations. However, manufacturers do not have enough comparative research when launching products that they claim to have advantages. Therefore, the study of these donors is extremely important for economy and consequently for sustainability, as it is fair in deposition. The effects of external factors were minimized by developing a test bench to measure output (g s$^{-1}$) of different GFs in planting rows using different spiral fertilizer spreaders (single and double) or fluted roller.

## Materials and methods

### Development of the test bench

The test bench Fig 1 evaluated the efficiency of three fertilizer spreaders (with one spiral roller, two spiral rollers, or a fluted roller) at three operating speeds (1.11, 1.94, and 2.77 m s$^{-1}$) using two NPK formulations (GF$_1$ 04-14-08 and GF$_2$ 04-30-10), totaling 18 treatments. For each treatment, 420 outflow measurements (g s$^{-1}$) were obtained, corresponding to 7,560 data points.

Electric drive via frequency inverter allowed precisely adjusting the speed of the 0.246 kW gear motor and drive the common axis of the dosing mechanism through a gear ratio. Operating speeds were determined based on the application of 300 kg ha$^{-1}$ GF in the spreader with a spiral roller and 250 kg ha$^{-1}$ in the spreader with a fluted roller, allowing both types of spreaders to work at the same rotation. Inter-row spacing was 0.50 m, resulting in a load of 15.0 and 12.5 grams of fertilizer per meter, respectively. This equipment has been parameterized to operate in the frequency of 1 to 60 hertz being activated by a linear pot of 5 KΩ, thus allowing to vary the working speed of the three dosers together, according to Table 1 and Fig 2.

In addition to the electric adjustment of the speeds, the test bench architecture allowed varying the longitudinal and transverse angles of the spreader using threaded bars sized to fit joints with an angle between −30˚ and +30˚ in both directions. The reservoirs at the upper end of the bench were connected to three types of spreaders, as follows: A. with one spiral roller and a pitch of 1 inch; B. with two spiral rollers and a pitch of ½ inch, and C. with a fluted roller, which worked with an eight-channel 6.9 cm$^3$-rotor arranged vertically.

### Data acquisition system

The distribution of GFs was measured with high precision and accuracy using a data acquisition system (DAS) in Arduino, a low-cost open-source software, at an acquisition frequency of 1 Hertz. This system was connected to three single-point load cell-type scales performing real-time measurements and collected 420 seconds of outflow data. The initial and final 30-second intervals were excluded because they corresponded to the period of flow stabilization, and collection was interrupted before the reservoir contents reached the final third. Thus, the average of the pulses was performed for each scale and, subsequently, a calibration curve was calculated Fig 3. With the equation found, the precision of the bench scale at 0.0011 grams per pulse was

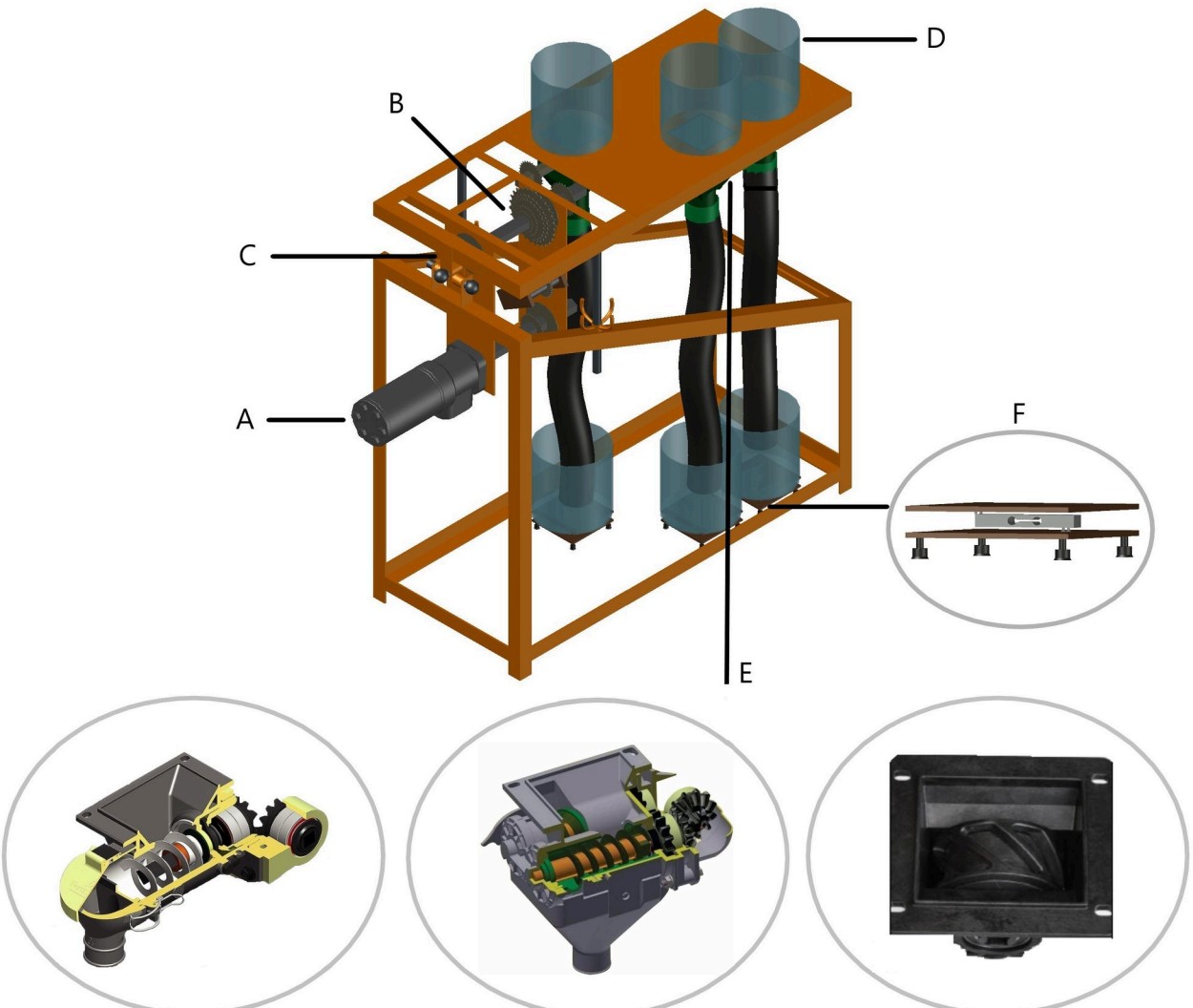

**Fig 1.** Design of the test bench: electric drive (A), transmission assembly (B), hinge system (C), reservoirs (D), dispensing mechanisms (E), and data acquisition system (F).

identified, that is, it is possible to count a single granule of fertilizer deposited on the scale in real time.

## Characterization of GFs

N-$P_2O_5$-$K_2O$ GFs were selected according to the products marketed in the region and the concentration of the formulations. Two tons of GFs were divided into two formulations: $GF_1$

**Table 1. Flow determination to simulate speeds.**

| km h-1 | m s-1 | g s-1 | g min-1 |
|---|---|---|---|
| 4,0 | 1,11 | 16,67 | 1.000 |
| 7,0 | 1,94 | 29,17 | 1.750 |
| 10,0 | 2,77 | 41,67 | 2.500 |

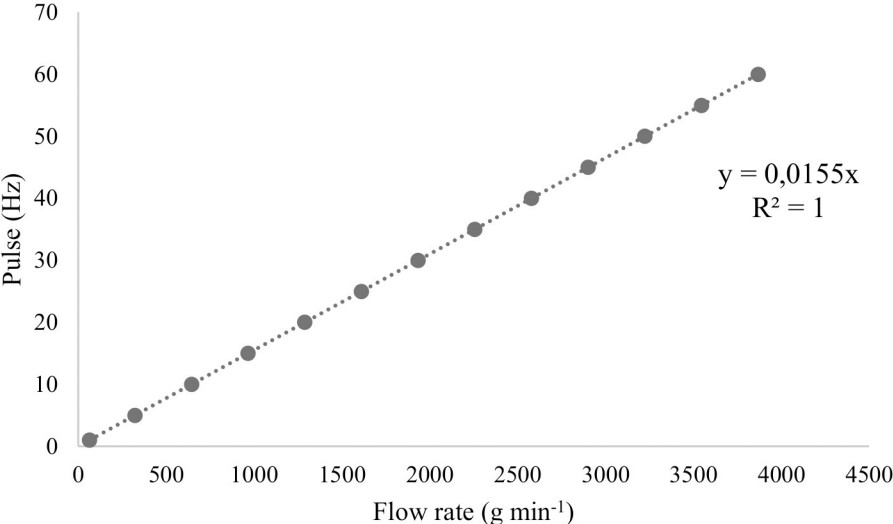

**Fig 2. Calibration curve as a function of flow rate.**

04-14-08 and $GF_2$ 04-30-10. Immediately after storage in a weather-protected area, GF particle size was determined using the following sieve set following the manufacturer's recommendations sit [6]: 4.0 mm (ABNT No. 05); 2.0 mm (ABNT No. 10); 1.0 mm (ABNT No. 18); 0.5 mm (ABNT No. 35), and the retained fraction. Particle density was estimated using Dalle Molle equipment, and fluidity was determined by the angle of repose using two kilograms of GF deposited in a rectangular glass vessel at a constant speed. The relative water content (percentage fresh weight) was determined according to the methodology proposed by sit [6] using an analytical balance. Only particle density was significantly different between the fertilizers Table 2. The angles of repose and water content were similar between $GF_1$ and $GF_2$.

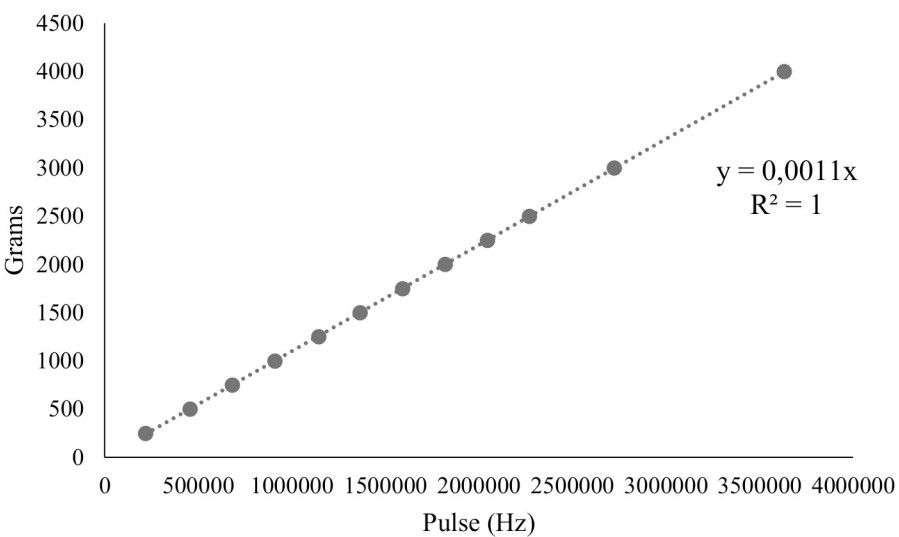

**Fig 3. Calibration curve for balance one, two and three.**

**Table 2. Density, angle of repose, and water content of granular fertilizers.**

| Ganular fertilizer (GF) | Density (g cm$^{-3}$) | Angle of repose (°) | Water content (g g$^{-1}$) |
|---|---|---|---|
| GF$_1$ 04-14-08 | 0.97 | 32.55 | 0.03 |
| GF$_2$ 04-30-10 | 0.95 | 33.69 | 0.03 |

**Table 3. Analysis of variance of the particle size of granular fertilizers.**

| Granular fertilizer (GF) | Mesh |
|---|---|
| GF$_1$ 04-14-08 | 19.90 |
| GF$_2$ 04-30-10 | 20.00 |
| Sieve (S) | |
| 0.0 mm | 0.00 |
| 0.5 mm | 0.25 |
| 1.0 mm | 14.87 |
| 2.0 mm | 81.38 |
| 4.0 mm | 3.50 |

**Table 4. Partial and accumulated amount of granular fertilizer passing through each sieve.**

| Sieve (S) | GF$_1$ 04-14-08 | | GF$_2$ 04-30-10 | |
|---|---|---|---|---|
| | Partial (%) | Accumulated (%) | Partial (%) | Accumulated (%) |
| 0.0 mm | 0.00 | 0.00 | 0.00 | 0.00 |
| 0.5 mm | 0.50 | 0.50 | 0.00 | 0.00 |
| 1.0 mm | 24.25 | 24.75 | 5.50 | 5.50 |
| 2.0 mm | 72.75 | 97.50 | 90.00 | 95.50 |
| 4.0 mm | 2.50 | 100.00 | 4.50 | 100.00 |

**Table 5. Descriptive statistics of the outflow of fertilizers using different fertilizer spreaders at a speed of 1.11 m s$^{-1}$.**

| Variable | Spiral Single | | Spiral Double | | Fluted Roller | |
|---|---|---|---|---|---|---|
| | GF$_1$ | GF$_2$ | GF$_1$ | GF$_2$ | GF$_1$ | GF$_2$ |
| Means | 16.52 | 15.70 | 18.48 | 15.31 | 14.03 | 13.86 |
| Median | 16.43 | 15.60 | 18.42 | 15.30 | 14.04 | 13.85 |
| Mode | 17.17 | 14.63 | 17.99 | 15.22 | 15.04 | 13.48 |
| Standard deviation | 1.02 | 1.03 | 1.30 | 1.10 | 0.99 | 1.11 |
| Amplitude | 5.69 | 6.43 | 7.52 | 6.19 | 6.46 | 6.73 |
| CV (%) | 5.87 | 6.66 | 7.60 | 6.92 | 7.17 | 7.92 |
| Asymmetry | 0.21 | 0.10 | 0.05 | 0.00 | −0.06 | −0.06 |
| Kurtosis | −0.27 | 0.04 | −0.10 | −0.25 | 0.19 | 0.05 |
| JB | 4.46 N | 0.74 N | 0.36 N | 1.08 N | 0.83 N | 0.32 N |

CV, coefficient of variation (%); JB, Jarque-Bera normality test (N, normal distribution; A: non-normal distribution at $p \leq 0.05$; AA, non-normal distribution at $p \leq 0.01$); GF, granular fertilizer.

Particle size was significantly different between the formulations Table 3. The 2.0 mm mesh retained the largest amount of $GF_1$ and $GF_2$, followed by the 1.0 mm mesh, whereas particle size was similar using the other meshes.

Granulometric characteristics were also analyzed Table 4. The 2.0 mm mesh increased the retention of $GF_1$ and $GF_2$ by 72.75% and 90%, respectively.

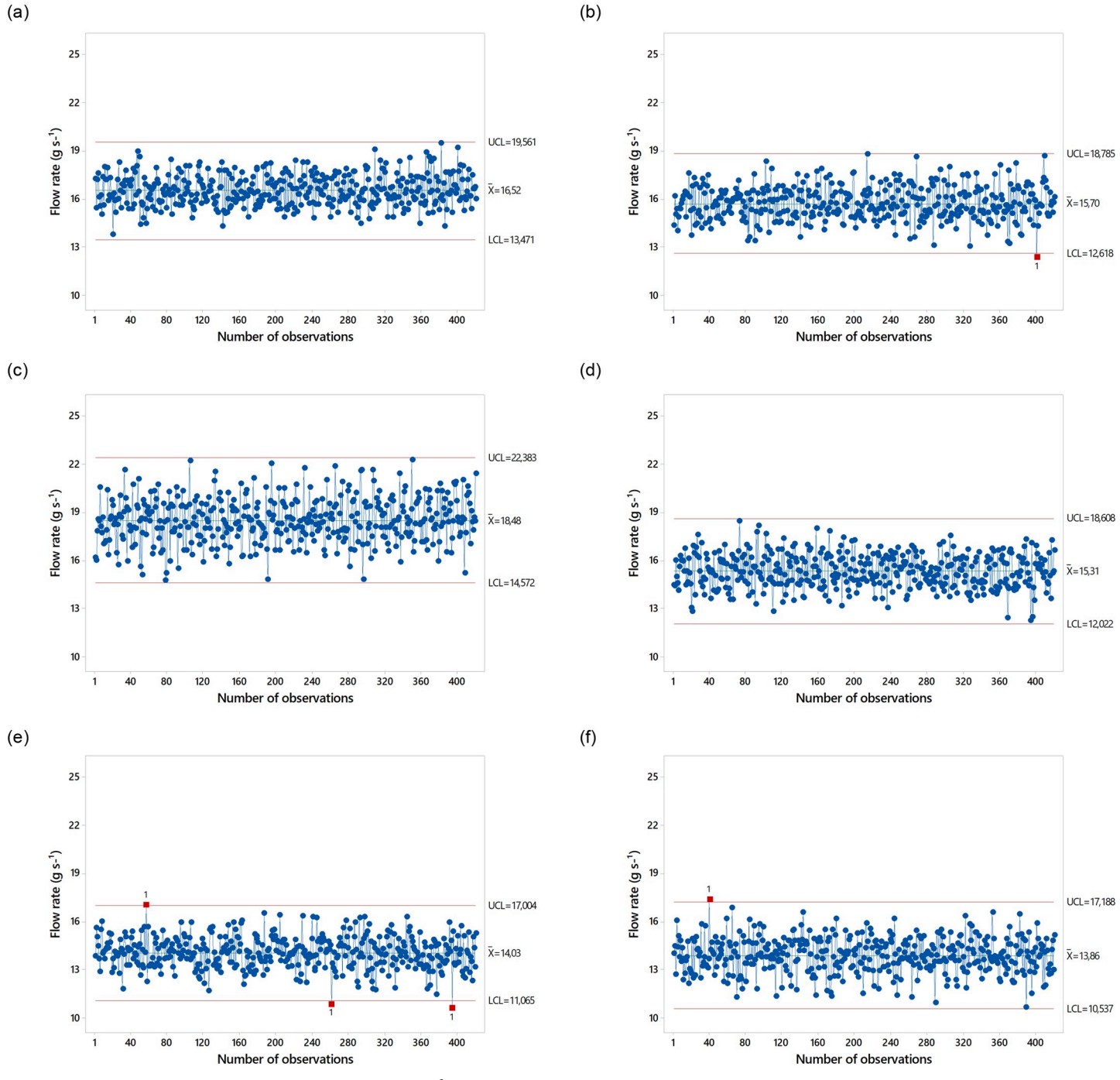

**Fig 4. Performance of fertilizer spreaders at a speed of 1.11 m s$^{-1}$.**

## Statistical analysis

A total of 7,560 outflow data points was subjected to descriptive analysis of the measures of central tendency (mean, median, and mode), dispersion (amplitude, standard deviation, and coefficient of variation), asymmetry, and kurtosis. The Jarque-Bera normality test was also performed sit [7]. Data were subjected to statistical process control, generating Shewhart control charts of averages for each spreader and velocity, allowing measuring the outflow rate and variability using lower and upper control limits sit [8]. These limits consider data variability due to uncontrollable circumstances and are calculated using standard deviations.

## Results and discussion

Central tendency was different between $GF_1$ and $GF_2$ in the three types of fertilizer spreaders. Spreader Spiral Single presented asymmetry between 0.21 and 0.10 for $GF_1$ and $GF_2$, suggesting that the curve was skewed to the right, given that the median was lower than the mean. Spreader Spiral Double had positive asymmetry for $GF_1$ but no asymmetry for $GF_2$. In spreader Fluted Roller, asymmetry was –0.06 for both GFs, indicating that the curve was skewed to the left. The descriptive statistics Table 5 of dispersion indicated the presence of kurtosis. For $GF_1$ in spreader Spiral Single, a kurtosis of –0.27 indicated the smallest number of outflow data points around the mean, i.e., a platykurtic distribution; however, for $GF_2$, kurtosis presented a leptokurtic distribution, with a high number of data points around the central tendency. Spreader Spiral Double presented a platykurtic distribution for both GFs, in contrast to spreader Fluted Roller for both formulations.

Data dispersion was low (CV≤10%). The average outflow values were higher for $GF_1$ in all three types of Spreader, which can be explained by the higher particle density of this formulation. The average outflow values are shown as Shewhart control charts in Fig 4. Spreader Spiral Single at the speed of 1.11 m s$^{-1}$ did not show any out-of-control events for $GF_1$. There was one out-of-control event for $GF_2$. For spreader Spiral Double, there were no out-of-control events for both formulations. For spreader Fluted Roller, there were out-of-control events for both formulations.

The descriptive statistics of outflow data in different spreaders at a speed of 1.94 m s$^{-1}$ are presented in Table 6. Spreader Spiral Single presented asymmetry between 0.13 for $GF_1$ and 1.41 for $GF_2$, indicating positive asymmetry. In contrast, asymmetry was negative in spreader

**Table 6. Descriptive statistics of outflow data of granular fertilizers using different fertilizer spreaders at a linear speed of 1.94 m s$^{-1}$.**

| Variable | Spiral Single | | Spiral Double | | Fluted Roller | |
|---|---|---|---|---|---|---|
| | $GF_1$ | $GF_2$ | $GF_1$ | $GF_2$ | $GF_1$ | $GF_2$ |
| Means | 29.70 | 28.08 | 32.37 | 26.15 | 24.60 | 24.18 |
| Median | 29.48 | 27.56 | 32.45 | 26.20 | 24.51 | 24.08 |
| Mode | 29.76 | 27.96 | 31.08 | 24.18 | 24.13 | 23.74 |
| Standard deviation | 2.08 | 1.89 | 1.89 | 1.52 | 1.44 | 1.37 |
| Amplitude | 10.55 | 17.09 | 14.32 | 10.81 | 9.79 | 8.33 |
| CV (%) | 6.23 | 7.29 | 7.46 | 6.40 | 5.79 | 5.67 |
| Asymmetry | 0.13 | 1.41 | -0.10 | -0.10 | 0.23 | 0.40 |
| Kurtosis | -0.40 | 5.48 | -0.16 | -0.30 | 0.73 | 0.42 |
| JB | 3.97 N | 664.64 N | 1.11 N | 2.31 N | 12.90 N | 14.46 N |

CV, coefficient of variation (%); JB, Jarque-Bera normality test (N, normal distribution; A: non-normal distribution at $p \leq 0.05$; AA, non-normal distribution at $p \leq 0.01$); GF, granular fertilizer.

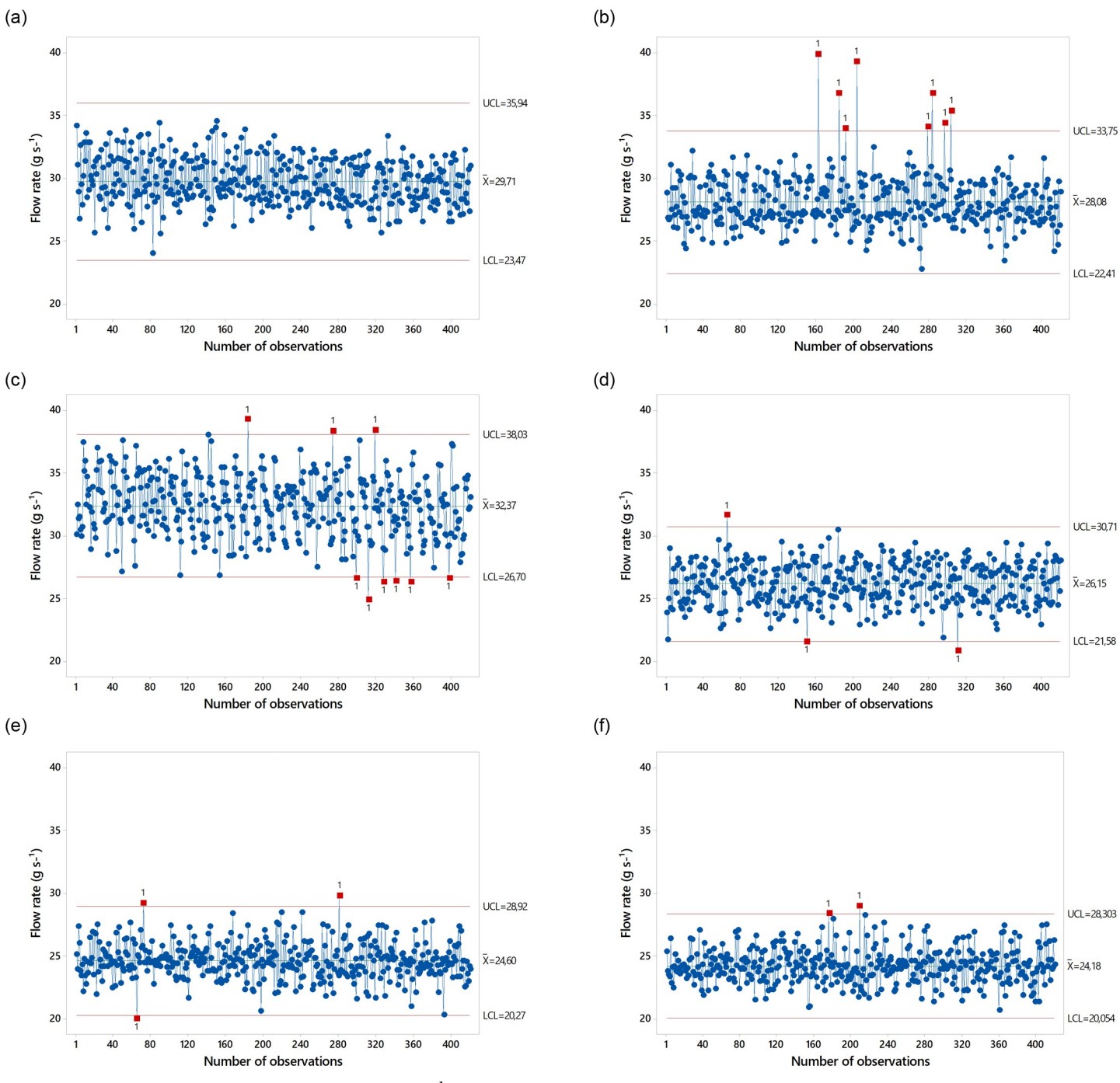

**Fig 5. Performance of fertilizer spreaders at a speed of 1.94 m s⁻¹.**

Spiral Double because the means were lower than the median. Asymmetry was positive in Fluted Roller, i.e., the curve was skewed to the right.

The descriptive statistics of dispersion evidenced the presence of kurtosis. For $GF_1$ in Spiral Single, kurtosis of –0.40 indicated a platykurtic distribution, in contrast to $GF_2$, whose data distribution was close to the mean (leptokurtic). Spiral Double presented a platykurtic

**Table 7. Descriptive statistics for outflow data of granular fertilizers using different fertilizer spreaders at a speed of 2.77 m s$^{-1}$.**

| Variable | Spiral Single | | Spiral Double | | Fluted Roller | |
|---|---|---|---|---|---|---|
| | GF$_1$ | GF$_2$ | GF$_1$ | GF$_2$ | GF$_1$ | GF$_2$ |
| Means | 43.99 | 39.15 | 47.50 | 36.71 | 35.04 | 32.63 |
| Median | 43.75 | 38.58 | 47.64 | 36.86 | 34.85 | 32.55 |
| Mode | 42.74 | 38.54 | 48.80 | 37.04 | 34.92 | 31.96 |
| Standard deviation | 2.62 | 2.50 | 3.62 | 1.91 | 1.86 | 1.56 |
| Amplitude | 18.28 | 19.42 | 19.27 | 16.50 | 12.28 | 10.79 |
| CV (%) | 6.35 | 6.49 | 7.18 | 6.67 | 5.48 | 5.06 |
| Asymmetry | 0.09 | 0.91 | 0.02 | -0.25 | 0.33 | 0.02 |
| Kurtosis | -0.01 | 2.47 | -0.38 | 0.37 | 0.40 | 0.74 |
| JB | 0.59 N | 164.95 N | 2.56 N | 6.94 N | 10.42 N | 9.50 N |

CV, coefficient of variation (%); JB, Jarque-Bera normality test (N, normal distribution; A: non-normal distribution at $p \leq 0.05$; AA, non-normal distribution at $p \leq 0.01$); GF, granular fertilizer.

distribution for both formulations. In Fluted Roller, the curve had a strong leptokurtic distribution for both formulations. Data dispersion was low (low CV values) between the spreaders and fertilizer formulations at a speed of 1.94 m s$^{-1}$. The average outflow values are shown as Shewhart control charts in Fig 5.

There were systematic deviations in the outflow of GF$_2$ in spreader Spiral Single at 1.94 m s$^{-1}$ but no systematic deviations in the outflow of GF$_1$. There were systematic deviations in the outflow of both formulations in spreaders Spiral Double and Fluted Roller at 1.94 m s$^{-1}$. The outflow rates between different spreaders and fertilizer formulations at a speed of 2.77 m s$^{-1}$ are presented in Table 7. Asymmetry was positive in spreaders Spiral Single and Fluted Roller for both GF$_1$ and GF$_2$, i.e., the mean was higher than the median, and the curve was skewed to the right. In contrast, asymmetry was negative in Spiral Double for both formulations, i.e., the median was higher than the mean.

GF$_1$ in spreader Spiral Single presented kurtosis of –0.01, confirming its platykurtic distribution and intermediate dispersion around the mean. GF$_2$ in Spiral Single presented a leptokurtic distribution, with a value of 2.47. Kurtosis was heterogeneous in spreader Spiral Double, with a platykurtic distribution for GF$_1$ and a leptokurtic distribution for GF$_2$. However, kurtosis was homogeneous in spreader Fluted Roller at a speed of 2.77 m s$^{-1}$ with a leptokurtic distribution for both formulations. The coefficients of variation, data dispersion, and standard deviation were small. However, the amplitudes of outflow variability were consistent with the higher data dispersion around the mean. It is of note that in both formulations, the averages were higher than the median, except for GF$_1$ and GF$_2$ in Spiral Double. The average outflow values are shown as Shewhart control charts in Fig 6.

Spreader Spiral Single at a speed of 2.77 m s$^{-1}$ presented systematic deviations in the outflow of both formulations. In spreader Spiral Double, there were no systematic deviations for GF$_1$, and dispersions were higher than the upper and lower ranges for GF$_2$. However, some data points extrapolated the established range for both formulations in spreader Fluted Roller. These results suggest that the speed of 1.11 m s$^{-1}$ presented the smallest systematic deviations, i.e., outflow variability values were lower for the three types of spreaders, flow data variability increased with increasing velocity. Despite the higher uniformity, GF$_2$ presented the highest number of systematic deviations at a speed of 1.94 and 2.77 m s$^{-1}$. The higher segregation of GF$_1$ led to higher deviations in outflow at a speed of 1.94 m s$^{-1}$ relative to the speed of 2.77 m s$^{-1}$. Spiral

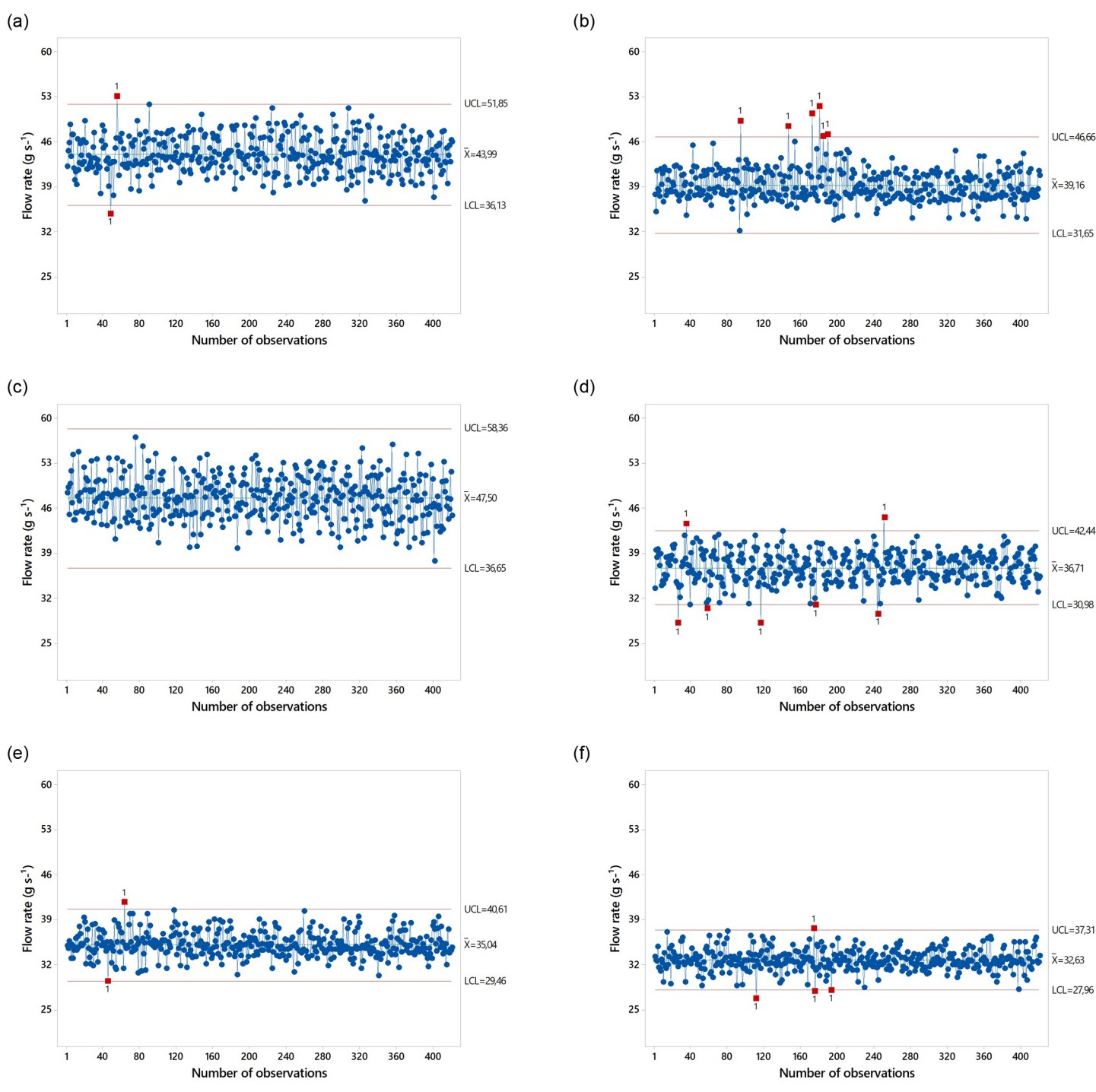

**Fig 6. Performance of fertilizer spreaders at a speed of 2.77 m s⁻¹.**

Double presented the highest deviations in the outflow of $GF_1$ at 1.94 m s⁻¹ and $GF_2$ at 2.77 m s⁻¹. Spiral Single presented the highest deviations in the outflow of $FG_2$ at 1.94 m s⁻¹ and both $GF_1$ and $GF_2$ at 2.77 m s⁻¹. However, spreader Fluted Roller had the best dispersion around the mean compared to Spiral Single and Spiral Double for all parameters.

## Conclusion

The bench and DAS allowed measuring the outflow of GFs in different spreaders. The segregation of granular fertilizers was different, and particle size variability was higher for $GF_1$ and lower for $FG_2$, affecting product outflow in the spreaders. The efficiency in the distribution of both formulations was higher at a speed of 1.11 m s$^{-1}$ and lower at 1.94 m s$^{-1}$. The fertilizer spreader with a fluted roller presented the highest uniformity in outflow rates at the evaluated speeds.

## Supporting information

**S1 Raw images.**
(PDF)

## Acknowledgments

This research was not funded by public, commercial, or nonprofit agencies.

## Author Contributions

**Conceptualization:** Gabriel Ganancini Zimmermann, Samir Paulo Jasper, Leonardo Leônidas Kmiecik, Lauro Strapasson Neto, Thiago Xavier da Silva, Yasser Alabi Oiole.

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
