## [Decision Letter · Decision Letter 0]

13 Aug 2020

PONE-D-20-09797

Development of a low-cost test bench to determine the distribution of granular fertilizers in planting rows using spiral roller, two spiral rollers and fluted roller

PLOS ONE

Dear Dr. Zimmermann,

Thank you for submitting your manuscript to PLOS ONE. After careful consideration, we feel that it has merit but does not fully meet PLOS ONE’s publication criteria as it currently stands. Therefore, we invite you to submit a revised version of the manuscript that addresses the points raised during the review process.

The authors need to follow the comments from both reviewers especially addressed the concerns raised by the 2nd reviewer. The title needs to be more focused with inclusion of important key words to understand the research study. Should add relevant information in the introduction to justify why the research was conducted in relation to the comparative efficiency of the conventional and new technologies. Also, focus on economics and other concerns.

We look forward to receiving your revised manuscript.

Kind regards,

Rafiq Islam, Ph.D.

Academic Editor

PLOS ONE

Journal Requirements:

Reviewers' comments:

Reviewer's Responses to Questions

**Comments to the Author**

1. Is the manuscript technically sound, and do the data support the conclusions?

Reviewer #1: Partly

Reviewer #2: Yes

2. Has the statistical analysis been performed appropriately and rigorously? 

Reviewer #1: Yes

Reviewer #2: Yes

3. Have the authors made all data underlying the findings in their manuscript fully available?

Reviewer #1: Yes

Reviewer #2: Yes

4. Is the manuscript presented in an intelligible fashion and written in standard English?

Reviewer #1: Yes

Reviewer #2: Yes

5. Review Comments to the Author

Reviewer #1: The title of the article does not correspond to the content. The article lacks data on the cost of the test bench and on the cost of the study.

In the presented figures 4a-4f and 5a-5f, it is necessary to expand the distribution of data along the Y-axis. This is necessary for greater clarity of the figures.

In the conclusions and annotations, it is necessary to indicate the cost of the test bench and how cheap it is to work in comparison with analogs.

Reviewer #2: The study sought to develop a low-cost test bench to determine the distribution of granular fertilizers in planting rows using spiral roller, two spiral rollers and fluted roller. The introductory section of the manuscript appears disjointed and incoherent. The aim of the study is conspicuously not made clear in the write-up.

The authors failed to compare the output flow rates of the developed test bench with already existing spreaders. What was the basis for comparison relative to the efficiency of this newly developed technology with already existing technologies?

The authors again failed to support their findings with what has been done elsewhere. The manuscript actually lacks the required literature review.

6. PLOS authors have the option to publish the peer review history of their article (what does this mean?). If published, this will include your full peer review and any attached files.

Reviewer #1: No

Reviewer #2: No

---

## [Author Response · Author response to Decision Letter 0]

28 Sep 2020

Review Comments to the Author

Reviewer #1: The title of the article does not correspond to the content. The article lacks data on the cost of the test bench and on the cost of the study.

In the presented figures 4a-4f and 5a-5f, it is necessary to expand the distribution of data along the Y-axis. This is necessary for greater clarity of the figures.

In the conclusions and annotations, it is necessary to indicate the cost of the test bench and how cheap it is to work in comparison with analogs.

A: The title of the manuscript was changed, removing the term low cost, and the economic direction of the development of the structure and its quotations. A new paragraph was added in the introduction, referring to the justification and importance of the work. The expansion of the data distribution of the commented figures, could not be attended for statistical and software reasons. As for the cost results in the conclusion, it will no longer be necessary due to the change in the title of the manuscript.

Reviewer #2: The study sought to develop a low-cost test bench to determine the distribution of granular fertilizers in planting rows using spiral roller, two spiral rollers and fluted roller. The introductory section of the manuscript appears disjointed and incoherent. The aim of the study is conspicuously not made clear in the write-up. The authors failed to compare the output flow rates of the developed test bench with already existing spreaders. What was the basis for comparison relative to the efficiency of this newly developed technology with already existing technologies? The authors again failed to support their findings with what has been done elsewhere. The manuscript actually lacks the required literature review.

A: The title of the manuscript was changed, removing the term low cost, and the economic direction of the development of the structure and its quotations. A new paragraph was added in the introduction, referring to the justification and importance of the work. The output flow rates were addressed before the descriptive statistics of the data and were not compared to other literature because there were no methods of evaluating the spreaders yet, so it was not possible to add new changes. This test bench was developed for this proposal to evaluate the regularity of spreader distribution, and the existing works do not meet the technical conditions for comparison.

---

## [Decision Letter · Decision Letter 1]

26 Nov 2020

Development of test bench to determine the distribution of granular fertilizers in planting rows using spiral roller, two spiral rollers and fluted roller

PONE-D-20-09797R1

Dear Dr. Zimmermann,

We’re pleased to inform you that your manuscript has been judged scientifically suitable for publication and will be formally accepted for publication once it meets all outstanding technical requirements.

Kind regards,

Mohammad Mehdi Rashidi

Academic Editor

PLOS ONE

Additional Editor Comments (optional):

The revised version could be accepted in the present form.

Editor

Reviewers' comments:

Reviewer's Responses to Questions

**Comments to the Author**

1. If the authors have adequately addressed your comments raised in a previous round of review and you feel that this manuscript is now acceptable for publication, you may indicate that here to bypass the “Comments to the Author” section, enter your conflict of interest statement in the “Confidential to Editor” section, and submit your "Accept" recommendation.

Reviewer #1: All comments have been addressed

Reviewer #2: All comments have been addressed

2. Is the manuscript technically sound, and do the data support the conclusions?

Reviewer #1: Yes

Reviewer #2: Yes

3. Has the statistical analysis been performed appropriately and rigorously? 

Reviewer #1: Yes

Reviewer #2: Yes

4. Have the authors made all data underlying the findings in their manuscript fully available?

Reviewer #1: Yes

Reviewer #2: Yes

5. Is the manuscript presented in an intelligible fashion and written in standard English?

Reviewer #1: Yes

Reviewer #2: Yes

6. Review Comments to the Author

Reviewer #1: (No Response)

Reviewer #2: General comments

This manuscript is an improved version of what was earlier on submitted. The comments raised by the reviewers in the previous review have been adequately addressed.

The development of test bench to determine the distribution of granular fertilizers will undoubtedly serve as a prototype that will improve the application and distribution of granular-based fertilizers for that will increase fertilizer use efficiency for enhanced ecosystem functions and services.

7. PLOS authors have the option to publish the peer review history of their article (what does this mean?). If published, this will include your full peer review and any attached files.

Reviewer #1: No

Reviewer #2: No

---

## [Editor Report · Acceptance letter]

3 Dec 2020

PONE-D-20-09797R1 

Development of test bench to determine the distribution of granular fertilizers in planting rows using spiral roller, two spiral rollers and fluted roller 

Dear Dr. Zimmermann:

I'm pleased to inform you that your manuscript has been deemed suitable for publication in PLOS ONE. Congratulations! Your manuscript is now with our production department. 

Kind regards, 

on behalf of

Professor Mohammad Mehdi Rashidi 

Academic Editor

PLOS ONE